# Relationship between Doping Prevalence and Socioeconomic Parameters: An Analysis by Sport Categories and World Areas

**DOI:** 10.3390/ijerph19159329

**Published:** 2022-07-30

**Authors:** José Luís Terreros, Pedro Manonelles, Daniel López-Plaza

**Affiliations:** 1CELAD, Spanish Commission of Anti-Doping Fight, 29016 Madrid, Spain; joseluis.terreros@aepsad.gob.es; 2International Sports Medicine Chair, Catholic University of San Antonio, 30107 Murcia, Spain; pmanonelles@ucam.edu

**Keywords:** anti-doping, sport, socioeconomic characteristics, prohibited substances, Olympics

## Abstract

Socioeconomic differences between countries, including corruption and doping scandals, have increased in the last few decades. The aims of the current investigation were to examine doping prevalence according to world areas and sport groups and its association with socioeconomic factors worldwide. The Anti-Doping Rule Violations (ADRVs) of 160 countries competing at 2016 Olympics were analyzed between 2013 and 2018. In addition, the relationship between doping prevalence and socioeconomic characteristics, including Human Development Index (HDI), Per Capita Income (PCI) and Corruption Index (CI), was investigated. Africa, Asia, and America were revealed to have a significantly lower doping prevalence than Europe and Oceania when observing the sum and the mean ADRV/10,000 inhabitants (*p* < 0.01). Strong to moderate correlations were identified between Corruption Index and ADRVs and HDI and ADRVs (*p* < 0.01). However, the number of Olympic athletes was positively associated with the ADRVs and the HDI (r = 0.663 and 0.424, respectively). In the comparison by sport groups, the Independent Recognized Sports (AIMS) showed significantly higher Adverse Analytical Findings (AAF) and ADRVs (*p* < 0.01) than Olympic and Recognized International Sports (ARISF). In conclusion, the results of the current study reveal doping prevalence differences between world areas and sport categories, identifying associations with socioeconomic characteristics of each country.

## 1. Introduction

The circumstances and conducts that constitute Anti-Doping Rule Violations (ADRVs) have been reviewed and broadened in the last few decades [1,2]. Currently, the World Anti-Doping Code includes among ADRVs, attempted use or possession of a prohibited substance or method; concealing and distribution of a forbidden substance by any member of the athlete staff; or evading sample collection [1].

In an attempt to promote and coordinate the anti-doping fight, the World Anti-Doping Agency (WADA) was created in 1999. For that purpose, WADA relies on several accredited laboratories around the world, where about 250,000 athletes’ samples are taken and analyzed every year. A summary of all analyses worldwide is published annually in the Anti-Doping Rule Violations report with a 2-year delay [3]. The examination of the last Analytical ADRVs from 2013 to 2018 only revealed a 0.55–0.75% prevalence of violations among the total number of samples analyzed worldwide [3]. However, prevalence research using randomized questionnaires revealed that up to 39% of elite athletes consumed prohibited substances intentionally in the last year [4,5]. Therefore, some investigations have questioned the efficiency of the system in the detection process and in the institutional strategy associated with the anti-doping fight [6,7,8].

Prior investigations have examined and compared doping use based on discipline [2,6], age and gender [9,10], or sport level [4,11]. Athletic success and financial gain have been identified as the most common reasons for performance-enhancing drug (PED) consumption among athletes [12,13,14]. Although most athletes consider doping use as dishonest and unhealthy [14,15], the attitudes towards doping use are complex and many other contributing factors play a role in this behavior [16]. Lazuras et al. [13] determined past doping behaviors and situational temptation as the main psychosocial predictors, while Dunn et al. [17] identified the overestimation of prevalence among other athletes (“false consensus effect”) as paramount for PED use [18]. In addition, parental pressure, social recognition and financial support have been identified as determining factors, especially in certain regions or cultures [9,14,19].

Despite the increasing efforts of WADA against doping practices worldwide in the last few decades, significant differences between countries and regions can be observed in ADRV values [1,3,20]. Some investigations and reports have identified doping risk zones associated with a high prevalence of certain substances compared to other regions [8,19]. In addition, other factors such as the number of controls per year and reported ADRV percentages significantly differ depending on the laboratory used [3,8,20]. It is well known that certain prohibited practices rely on the acquiescence or even the explicit support of official institutions such as sports federations or governments [12,21]. This is the case for the German Democratic Republic (GDR) in the 1970s; China in the 1980s and 1990s; the International Association of Athletics Federation (IAAF); and the governments of Russia, Romania and Ukraine and National Anti-doping Organizations in the last decade [12,22].

Currently, the differences between countries in terms of social wellness and wealth are constantly increasing [23,24,25,26], and fraudulent conducts are more commonly accepted in poorer regions [27,28]. Furthermore, culture and educational programs play a key role in the prevention of PED use [7,14,29]. Since doping use can be considered a form of corruption, it is hypothesized that countries and regions with higher corruption rates and lower social and economic levels would also present significant greater ADRVs levels. Therefore, the aims of the current investigation were: (a) to investigate differences in ADRVs between world areas; (b) to determine the relationship between socioeconomic factors and doping prevalence; and (c) to compare the ADRVs-related differences between group of sports.

## 2. Materials and Methods

### 2.1. Analysis by World Areas

A total of 160 countries were selected for this study. The inclusion criteria were: (a) participation in the Olympic Games of 2016 in Rio de Janeiro and (b) anti-doping data availability in the annual Anti-Doping Rule Violations (ADRVs) report [3]. The socio-economic parameters were obtained for each country between 2013 and 2018 and included Human Development Index (HDI), Per Capita Income (PCI) and Corruption Index (CI) [24,25,26,30,31]. However, ADRVs were analyzed in the same period of time, taking into consideration country population and the number of participants in the 2016 Olympics [32]. Countries were classified by continents except for Europe which was subdivided into South Europe and North-Central Europe because of the traditionally different types of sports that are more popular in each area [19].

### 2.2. Analysis by Discipline

The cases of Adverse Analytical Findings (AAFs) were investigated between 2013 and 2018 by sport discipline. According to International Olympic Committee (IOC) [33], the sports were classified in three categories: Summer and Winter Olympic Sports (ASOIF—AIOWF), such as athletics or ski jumping; Recognized International Sports (ARISF), such as motor sports or sumo; and Independent Recognized Sports (AIMS), such as bodybuilding or kickboxing. Subsequently, each type of sport was analyzed based on the nature of the AAF: medical reason, no case to answer, no sanction, pending and ADRV.

### 2.3. Statistical Analysis

All statistical analyses were performed using the Statistical Package for the Social Sciences (SPSS) v24.0 (SPSS Inc., Chicago, IL, USA). Measures of homogeneity and spread are reported as mean and standard deviation (SD), while the level of significance was set as *p* < 0.05. Homogeneity of variance and normality of the distribution hypothesis were analyzed using Levene’s test and the Kolmogorov–Smirnov test, respectively. The one-way analysis of variance (ANOVA) test was performed to investigate: (a) the ADRVs by world areas (five levels: Africa, Asia, Oceania, America, South Europe, and North-Central Europe); and (b) the AAF by sport category (three levels: ASOIF—AIOWF, ARISF and AIMS), if no violations of the assumptions of normality and homogeneity were found. When one-way ANOVA analysis revealed significant differences, post hoc Bonferroni tests were conducted to identify the differences between groups. Kruskal–Wallis and post hoc Mann–Whitney tests with Bonferroni corrections (0.05/3) were conducted when the normality supposition of data was rejected, and significant differences were determined. To determine the interrelationships between the ADRVs by areas and the socioeconomic parameter, the Pearson’s correlation coefficient (r) was used, while Spearman’s correlation coefficient (r_s_) was used when the assumptions of normality were violated. Additionally, for the identification of the socioeconomic predictors of ADRV-As prevalence, a stepwise multiple linear regression analysis was performed excluding the non-significant variables observed in the previous linear correlation.

## 3. Results

The differences in ADRV-As prevalence between world areas are presented in Table 1. Analyzing the sum of ADRV-As/100,000 hab, Africa was revealed to have significantly lower values than Australia and Oceania, and all of Europe (*p* < 0.01). Likewise, significant differences were identified between America and Northern and Central Europe (*p* < 0.01). Regarding the mean ADRV-As/100,000 hab, significantly higher values were observed in Southern Europe with respect to Africa and Asia (*p* < 0.01). However, when the mean ADRV-As were analyzed with respect to the number of Olympic athletes in Rio de Janeiro 2016, no significant differences were found between regions.

Table 2 shows the linear relationship between the ADRV-As prevalence variables and the main socioeconomic indexes of each country worldwide (Appendix A). Both the sum and the mean ADRVs per 100,000 inhabitants are positively and significantly associated with the HDI and the Corruption Index of each country, with r values greater than 0.47 (*p* < 0.01). Similarly, in Figure 1, a high correlation was observed between the number of Olympians participating in the Rio 2016 Olympics and the mean AVDR in the period 2013–2018 (r = 0.663; *p* < 0.01), and between the number of Rio 2016 Olympians and the HDI. Although low, other significant associations were identified between sum ADRV/100,000 inhabitants and PCI and between the number of Rio 2016 Olympians and the Corruption Index (r > 0.35).

The stepwise linear regression equations that identify determining socioeconomic factors that predict ADRV-As are presented in Table 3. HDI, Corruption Index and the N° of Olympic athletes in Rio 2016 significantly contributed to predict ƩADRV-As/100,000 inhabitants and mean ADRV-As/100,000 inhabitants. However, total mean ADRV-As was only predicted by the N° of Olympic athletes in Rio 2016.

The results of doping controls per 1000 inhabitants and their comparison between categories are shown in Table 4. The analysis of AAF reveals significant differences between all groups of sports, with the highest values identified in AIMS sports, followed by ARISF and ASOIF (*p* < 0.05). The same trend can be observed in the pending cases and ADRV-As, where significant differences were determined between AIMS and ARISF sports and between AIMS and ASOIF (*p* < 0.05). Despite not revealing significant differences, a similar trend can be observed in the No case, with AIMS sports revealing the highest values.

## 4. Discussion

To the best of our knowledge, this is the first study to analyze ADRV cases regarding geographical areas and doping prevalence based on the socioeconomic level of the population. In accordance with the hypothesis, the level of corruption of each country is associated with ADRV prevalence whereas the HID supposition was not confirmed. Additionally, the sport group investigation revealed the significantly higher ADRV and AAF incidence of AIMS sports compared to ASOIF-AIOWF and ARISF disciplines. These results provide normative data regarding the importance of social and economic level of countries in doping prevalence and the world areas where it is necessary to reinforce the anti-doping fight.

The number of anti-doping tests performed worldwide is not equally distributed by the number of elite athletes, country population or number of professional sport licenses [3,20]. This is especially significant in the out-of-competition tests with no relationship between the degree of traditional sport success of the countries and the frequency of tests [8]. Similarly, Morente et. al [14] an observed unequal distribution of doping investigations between countries and geographical areas. Thus, the present investigation analyzed the ADRV values based on the population of well-defined regions. In the present study, all regions of Europe revealed significantly greater ADRV prevalence per 100,000 inhabitants compared with Africa, America, or Asia *(p* < 0.01), which might be related to the higher possibilities of prohibited substance use [16], the higher number of anti-doping tests undertaken in those areas and the more numerous sport practitioners there [20]. In agreement with these findings, no differences were observed when examining the ADRVs according to the number of Olympians in any region of the world. Although the investigation of doping risk zones is scarce, previous studies and reports have identified a trend in the detection of certain substances according to the location of the laboratories and the origin of the samples [2,3]. Manonelles et al. [2,19] reported an increase in the use of anabolic steroids in countries of Middle and Eastern Europe and the use of hormones in Mediterranean countries. Therefore, world region is a determinant factor in doping prevalence and the type of substances used by athletes. Perhaps, the anti-doping fight should focus on the detection of substances traditionally related to particular sports according to word areas and ultimately the prevention of their use.

Theoretically, competitive sport, particularly at the highest level, implies the respect of rules and fair play in the pursuit of an honest and fair competition [34,35]. Any attempt to overcome these rules might be considered fraudulent and unethical. Similarly, the significant association between ADRVs and Corruption Index identified in the present study, also as a predictor, confirms that the countries with more fraudulent practices were also the ones with higher levels of doping use. Therefore, targeted educational programs and overall strategies for doping prevention should be implemented, especially in these countries [7].

Regarding socioeconomic parameters, the current investigation revealed HDI as a predictor for ADRVs. HDI considers not only Per Capita Income but also life expectancy and level of education of country population. These findings agree with human development theories such as Maslow’s hierarchy of needs [36,37]. In countries where necessities are difficult to fulfil, PED access or sport success seems secondary, whereas in more developed countries the pursuit of esteem and self-actualization play a more important role. Prior investigations have determined financial gain and social recognition as the main reasons for doping use that might be considered forms of self-fulfillment [9,14,19,38]. In addition, when analyzing doping prevalence based on sport level, athletes’ doping intentions seem to increase according to sport level, observing greater values in the more successful athletes and in senior categories [4,11,18]. In accordance, the current investigation identified significant and positive associations between the number of Río 2016 Olympians and ADRVs values.

The first case of doping promoted by a country took place in the extinct German Democratic Republic in the 1970s, in which a doping system was created and controlled by the political police of the country (STASI). The main substance was the anabolic Turinabol, produced in a state factory where hundreds of doctors and scientists were implicated, and near 20,000 athletes were also involved [38]. Another country that was implicated in massive and apparently consensual doping practices was the People’s Republic of China in the 1980s and 1990s with systematic doping practices in athletics [12]. Further institutional infractions on doping occurred at the IAAF in 2015 in which a system of corruption and extortion was favored in collusion with the Russian Athletics Federation, and responsible Russian sports organizations becoming the first breakdown of anti-doping procedures related to the Athlete Biological Passport (“ABP”) [21]. The latest state doping scandal recently occurred in Russia, in which an institutional conspiracy of the Ministry of Sport and its infrastructure has been demonstrated. Reportedly, RUSADA (Russian National Anti-Doping Agency), CSP (Center for Sports Preparation of Russian National Teams), and the Moscow Doping Control Laboratory with FSB (Russian Federal Security Service) aimed at manipulating doping tests in the WADA computer system [22]

Previous investigations on PED use have determined significant differences in doping prevalence when comparing sport disciplines and categories [6,14,15]. Apparently, doping prevalence among individual and power sports athletes is higher than in team or motor skill sports, but they are also more likely to be tested [13,39,40]. According to the findings of the current study, Mottram et al. [41] observed that athletes of traditional Olympic sports such as cycling, athletics or powerlifting understood doping rules and procedures better than athletes of other disciplines. Perhaps, the greater number of doping tests undertaken by ASOIF—AIOWF athletes compared with ARISF, but especially AIMS athletes (ADRV report), might be related to the significant differences observed here in AAFs and ADRVs. Furthermore, increasing the frequency of doping tests has previously been identified as an effective strategy in the anti-doping fight [3,8]. These results suggest that athletes in sports with less anti-doping control, and those less traditionally attached to Olympic values and idiosyncrasy, were also more likely to use banned drugs.

The present investigation has some limitations that should be taken into consideration. In the comparisons between geographical areas, ADRV parameters were normalized for 100,000 inhabitants. Since not all countries and regions had the same percentage of athletes among their population, the analysis would have been more precise if it were based on the number of sport licenses. However, not all countries require licenses to train and compete, making the control of the number of athletes a difficult task. Another limitation of the present investigation is related to the outdated information on doping prevalence because of the two-year delay in the ADRV reports annually published by WADA. In addition, future research might use newer and more complete corruption measures [42]. 

## 5. Conclusions

Overall, the findings of the present investigation revealed a high doping prevalence in all regions of Europe, Australia, and Oceania. However, in the sport-related comparison, traditional Olympic sports presented lower ADRV values than AIMS and ASOIF sports. In addition, the influence of socioeconomic parameters on doping use was identified since more economically developed and corrupted countries were associated with larger doping prevalence values. Hopefully, these results might contribute not only to providing normative data but also to identifying the risk factors and geographical areas where anti-doping fight and prevention needs to be reinforced according to sport discipline.

## Figures and Tables

**Figure 1 ijerph-19-09329-f001:**
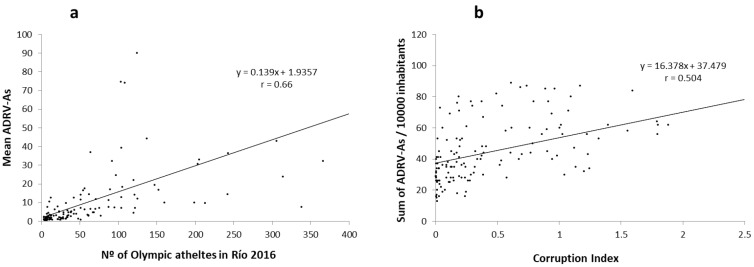
Relationship between (**a**) ADRV-As and the number of Olympic athletes taking part in Río 2016 Olympic Games of each country; (**b**) sum of ADRV-As and the Corruption Index of each country.

**Table 1 ijerph-19-09329-t001:** Prevalence of ADRV-As by world areas between 2013 and 2018.

Variable	Area	Mean ± SD	95% CI
ƩADRV-As/100,000 inhab	Africa	0.10 ± 0.11 *^†§^	0.06–0.15
	Asia	0.27 ± 0.36 ^†§^	0.16–0.38
	Australia and Oceania	1.02 ± 0.75	0.33–1.71
	America	0.53 ± 0.58 ^§^	0.32–0.73
	South Europe	1.01 ± 0.88	0.52–1.49
	North-Central Europe	1.06 ± 0.81	0.74–1.37
Mean ADRV-As/100,000 inhab	Africa	0.03 ± 0.04 ^†^	0.02–0.04
	Asia	0.05 ± 0.06 ^†^	0.03–0.07
	Australia and Oceania	0.43 ± 0.42	0.04–0.82
	America	0.18 ± 0.26	0.09–0.27
	South Europe	0.39 ± 0.73	−0.02–0.80
	North-Central Europe	0.18 ± 0.15	0.13–0.24
Ratio mean ADRV-As/N° Olympic athletes in Rio 2016	Asia	0.26 ± 0.25	0.18–0.34
	Australia and Oceania	0.10 ± 0.07	0.01–0.19
	America	0.19 ± 0.20	0.11–0.26
	South Europe	0.18 ± 0.12	0.11–0.25
	North-Central Europe	0.18 ± 0.14	0.12–0.23
Total mean ADRV-As	Africa	4.74 ± 8.99	1.38–8.10
	Asia	11.23 ± 19.58	5.13–17.33
	Australia and Oceania	8.02 ± 14.21	−5.12–21.17
	America	8.37 ± 14.09	3.38–13.37
	South Europe	16.36 ± 33.99	−2.47–35.18
	North-Central Europe	23.04 ± 31.88	10.68–35.40

Ʃ: Sum; ADRV-As: analytical anti-doping rule violation. * Significant differences (*p* < 0.01) with Australia and Oceania. ^†^ Significant differences (*p* < 0.01) with South Europe. ^§^ Significant differences (*p* < 0.01) with North-Central Europe.

**Table 2 ijerph-19-09329-t002:** Relationship between ADRV-As prevalence variables and socioeconomic indexes between 2013 and 2018.

	Ʃ ADRV-As/100,000 Inhab	Mean ADRV-As/100,000 Inhab	N° of Olympic Athletes in Rio 2016	Total Mean ADRV-As
N° of Olympic athletes in Rio 2016	0.010	−0.098	1	
Total mean ADRVs	0.150	−0.036	0.663 *	1
HDI	0.497 *	0.353 *	0.424 *	0.265 *
PCI	0.305 *	0.164	0.285 *	0.151
Corruption Index	0.504 *	0.474 *	0.384 *	0.152

Ʃ: Sum; ADRV-As: analytical anti-doping rule violation; HDI: Human Development Index; PCI: Per Capita Income; * significant relationship (*p* < 0.01).

**Table 3 ijerph-19-09329-t003:** Regression equations to predict ADRV-as based on socioeconomic parameters.

Variable	Equation	R^2^	*SEE*
Ʃ ADRV-As/100,000 inhab	−1.23 + (1.87 × HDI) − (0.01 × N° of Olympic athletes in Rio 2016) + (0.09 × Corruption Index) *	0.33	0.05
Mean ADRV-As/100,000 inhab	−0.207 + (0.002 × Corruption Index) − (0.001 × N° of Olympic athletes in Rio 2016) + (0.319 × HDI) *	0.32	0.10
Total mean ADRV-As	2.248 + (0.143 × N° of Olympic athletes in Rio 2016) *	0.43	17.97

Ʃ: Sum; ADRV-As: analytical anti-doping rule violation; HDI: Human Development Index; * significant relationship (*p* < 0.01).

**Table 4 ijerph-19-09329-t004:** Number of cases per 1000 anti-doping tests based on sport groups between 2013 and 2018.

	ASOIF—AIOWF		ARISF		AIMS		Post-hoc
	Mean ± SD	95% CI	Mean ± SD	95% CI	Mean ± SD	95% CI	*p*-Value
AAF	7.64 ± 4.41	6.17–9.11	21.25 ± 18.10 *	15.30–27.20	49.66 ± 36.48 ^†^^§^	31.52–67.80	<0.001
Medical reason	1.26 ± 1.14	0.88–1.64	2.97 ± 4.08	1.63–4.31	1.78 ± 3.49	0.04–3.51	0.058
No case	0.68 ± 0.58	0.49–0.88	0.92 ± 1.39	0.46–1.38	2.45 ± 4.91	0.01–4.89	0.029
No sanction	0.81 ± 0.44	0.66–0.96	1.16 ± 2.47	0.34–1.97	0.77 ± 1.25	0.15–1.39	0.600
Pending	0.68 ± 0.75	0.43–0.94	5.93 ± 8.98	2.97–8.88	14.23 ± 14.10 ^†^^§^	7.22–21.24	<0.001
ADRV-As	4.13 ± 3.17	3.08–5.19	10.45 ± 10.90	6.87–14.03	30.43 ± 28.65 ^†^^§^	16.19–44.68	<0.001

AAF: adverse analytical finding; ADRV-As: analytical anti-doping rule violation; * significant difference *(p* < 0.05) between ARISF and ASOIF; † significant difference *(p* < 0.05) between AIMS and ASOIF; § significant difference *(p* < 0.05) between AIMS and ARISF.

## Data Availability

All data presented in the current study are available on request.

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
