# Peer review of "Relationship between Doping Prevalence and Socioeconomic Parameters: An Analysis by Sport Categories and World Areas"

_ijerph, 2022, doi:10.3390/ijerph19159329_

Round 1

Reviewer 1 Report

Thank you for the opportunity to review this manuscript, it is very interesting manuscript with an intriguing topic.

Overall, you had a good paper about the relationship between doping prevalence and socioeconomic parameters. The Anti-Doping Rule Violations (ADRVs) of 168 countries were analyzed during five years between 2013 and 2018, and several variables and areas were investigated.

-          Your introduction, material, and methods sounded good. The results section was well organized and can be improved by using graphs if possible.

-          Lines 38: I think you mean 0.55-0.75%.

-          If possible, to list the countries which have not been investigated as an appendix.

Author Response

First of all we would like to thank you for taking time to review this research. We will provide a response point-by-point

  • Your introduction, material, and methods sounded good. The results section was well organized and can be improved by using graphs if possible.
    • A graph has been included in figure 1

  • Lines 38: I think you mean 55-0.75%.
    • Yes, it has been modified. Thanks

  • If possible, to list the countries which have not been investigated as an appendix.
    • An appendix has been included

Reviewer 2 Report

This is a very well written manuscript. Below are my comments

Introduction

Well written introduction with well defined aims. 

Methods

Considering the literature that's popular, I think the choices of measures were appropriate.

For future work, you may want to consider the work by Yeun Yeun Ang on the corruption index that she has created. It is more nuanced and may be of interest for a study such as the one that you have performed.

Based on your question of interest, why did you not use a regression?

Results:

For table 1, the column on the very right side is quite confusing. Can you please just use something along the lines of < Australia and Oceana, etc, instead of writing the p-value in there as well. I think that last column should be named post-hoc since that is where you present post-hoc results.

I don't think an uncontrolled analysis truly gives a full picture of the results. Please run multi-variate regression analyses as it will give you a better idea of the relationship between these variables. 

When I read your written results Table 3 makes perfect sense however, Table 3 should be self-explanatory and it's not. I would perhaps add another column for post-hoc. In Table 3, instead of "and" the authors of written "y". 

Discussion

I cannot fully judge the discussion since I believe that Table 2 should be a series of regressions instead of bivariate Pearson correlations. 

Author Response

First of all, we would like to thank you for taking time to review this research and the comments to improve it. We provide below a response point-by-point

1) For future work, you may want to consider the work by Yeun Yeun Ang on the corruption index that she has created. It is more nuanced and may be of interest for a study such as the one that you have performed.

We will take into consideration the investigations of Yeun Yeun Ang for future research on the field

2) Based on your question of interest, why did you not use a regression?

A multilinear regression analysis has been included in the results section

3) For table 1, the column on the very right side is quite confusing. Can you please just use something along the lines of < Australia and Oceana, etc, instead of writing the p-value in there as well. I think that last column should be named post-hoc since that is where you present post-hoc results.

Last column has been removed and significant p values have been replaced by symbols for a better understanding to readers

4) I don't think an uncontrolled analysis truly gives a full picture of the results. Please run multi-variate regression analyses as it will give you a better idea of the relationship between these variables.

A multilinear regression analysis has been included in the results section

5) When I read your written results Table 3 makes perfect sense however, Table 3 should be self-explanatory and it's not. I would perhaps add another column for post-hoc.

An additional column has been added. We hope it is what the reviewer meant

6) In Table 3, instead of "and" the authors of written "y".

It has been corrected

Reviewer 3 Report

The article analyses a relevant topic and is written in a simple, fluid manner but with appropriate methodology. The objective is to examine doping prevalence according to world areas and sport groups. It also looks at the association of these items with socioeconomic factors. I do not propose any changes or additions to the text. I would like you to revise the text between lines 207 and 222. Although relevant facts, in my opinion, they do not identify with the discussion points of the study.

line 38: 2019 or 2018(?)

Revise the abstract - with so many acronyms it becomes difficult to read and understand the gist.

Study limitations are well identified, given that the number of athletes per inhabitant would be important data to include in this study...

In the references, you must anticipate a number. Otherwise, references are all wrong (delete the blank line in reference 1.).

 It is an interesting study, and it can be accepted for publication in its current form.

Author Response

First of all, we would like to thank you for taking time to review this research and the comments to improve it. We provide below a response point-by-point

1) I would like you to revise the text between lines 207 and 222. Although relevant facts, in my opinion, they do not identify with the discussion points of the study.

The beginning of the paragraph has been reviewed and modified to address the point of the investigation. However, the main idea of the paragraph has been maintained since the authors consider that it provides a potential explanation of the HDI findings regarding doping.

2) line 38: 2019 or 2018(?)

Fixed

3) Revise the abstract - with so many acronyms it becomes difficult to read and understand the gist.

The number of acronyms has been reduced for an easier reading

4) Study limitations are well identified, given that the number of athletes per inhabitant would be important data to include in this study...

There is a real problem to identify the number of professional/elite athletes in each country since most of them do not use federative license or any other type of control to participate in competitions. In future research the authors we will try to address the problem by contacting each National Olympic Committee to try to get the registered athletes

5) In the references, you must anticipate a number. Otherwise, references are all wrong (delete the blank line in reference 1.).

Fixed

Round 2

Reviewer 2 Report

I'd like to thank the authors for addressing my concerns. I would just make it a priority to mention in your limitations that you did not use this newer meausre of corrpution (the one designed by Ang) which may be more robust.